# Macromolecule Translocation across the Intestinal Mucosa of HIV-Infected Patients by Transcytosis and through Apoptotic Leaks

**DOI:** 10.3390/cells12141887

**Published:** 2023-07-18

**Authors:** Susanne M. Krug, Carolin Grünhagen, Kristina Allers, Christian Bojarski, Joachim Seybold, Thomas Schneider, Jörg-Dieter Schulzke, Hans-Jörg Epple

**Affiliations:** 1Clinical Physiology/Nutritional Medicine, Charité—Universitätsmedizin Berlin, 12203 Berlin, Germany; 2Department of Gastroenterology, Rheumatology and Infectious Diseases, Charité—Universitätsmedizin Berlin, Campus Benjamin Franklin, 12203 Berlin, Germany; 3Antibiotic Stewardship Team, Medical Directorate, Charité—Universitätsmedizin Berlin, 10117 Berlin, Germany

**Keywords:** HIV, intestine, transcytosis, apoptosis, macromolecule passage, tight junction

## Abstract

Based on indirect evidence, increased mucosal translocation of gut-derived microbial macromolecules has been proposed as an important pathomechanism in HIV infection. Here, we quantified macromolecule translocation across intestinal mucosa from treatment-naive HIV-infected patients, HIV-infected patients treated by combination antiretroviral therapy, and HIV-negative controls and analyzed the translocation pathways involved. Macromolecule permeability was quantified by FITC-Dextran 4000 (FD4) and horseradish peroxidase (HRP) flux measurements. Translocation pathways were addressed using cold inhibition experiments. Tight junction proteins were characterized by immunoblotting. Epithelial apoptosis was quantified and translocation pathways were further characterized by flux studies in T84 cell monolayers using inducers and inhibitors of apoptosis and endocytosis. In duodenal mucosa of untreated but not treated HIV-infected patients, FD4 and HRP permeabilities were more than a 4-fold increase compared to the HIV-negative controls. Duodenal macromolecule permeability was partially temperature-dependent and associated with epithelial apoptosis without altered expression of the analyzed tight junction proteins. In T84 monolayers, apoptosis induction increased, and both apoptosis and endocytosis inhibitors reduced macromolecule permeability. Using quantitative analysis, we demonstrate the increased macromolecule permeability of the intestinal mucosa in untreated HIV-infected patients. Combining structural and mechanistic studies, we identified two pathways of increased macromolecule translocation in HIV infection: transcytosis and passage through apoptotic leaks.

## 1. Introduction

Raised systemic levels of gut-derived microbial macromolecules such as lipopolysaccharides (LPS), bacterial DNA, and β-D-glucan are drivers of immune activation and comorbidity in untreated HIV infection [1,2,3,4,5,6]. It was suggested that they resulted from the increased translocation across the epithelial barrier of the gut mucosa [2,7,8,9] and various structural and molecular alterations of the intestinal mucosa were proposed as correlates of a barrier defect to macromolecule permeation [10,11,12,13,14,15]. Because of these findings, it has been widely accepted that pathological macromolecule translocation represents a key mechanism in the pathogenesis of HIV infection [1,2,7]. However, despite its assumed prominent role in the disease, macromolecule translocation across the intestinal mucosa of HIV-infected patients has not been quantified so far. Also, whether or not the aforementioned molecular alterations of the intestinal mucosa observed in HIV-infected patients are functionally relevant for mucosal macromolecule permeability is unclear. To this end, the conception of increased mucosal translocation in HIV-infected patients is presently based only on indirect evidence. The main argument for this conception was derived from the consistent finding of increased plasma levels of LPS and other gut-derived microbial macromolecules in HIV-infected patients. However, these could also result from mechanisms independent of the intestinal mucosal barrier. For example, in a mouse model of HIV infection, increased serum levels of LPS were caused by decreased serum clearance owing to a defective macrophage function [16]. Therefore, it is presently unknown whether the increased systemic levels of microbial macromolecules found in untreated HIV-infected patients are secondary to enhanced translocation across the intestinal mucosa or rather caused by immunological mechanisms unrelated to the mucosal barrier function.

The primary goal of the present study was to answer the question of whether or not an untreated HIV infection is associated with an intestinal mucosal barrier defect to macromolecules. As we did find an increased macromolecule permeability of the intestinal mucosa in HIV-infected patients, our secondary goal was to characterize the translocation pathways involved. In general, macromolecule passage across the intestinal epithelium can occur either via a transcellular or a paracellular route, i.e., crossing or bypassing the enterocytes. It has been a long-held notion that transcellular passage is the main pathway of macromolecules translocation because the permeability of the paracellular pathway would not allow for the passage of molecules with a diameter of >3.6 Å or molecular weight of >10 kDa [17,18]. During the last decade, this view has been challenged. It was acknowledged that macromolecule passage can occur also in areas of epithelial damage [19]. Furthermore, the involvement of occludin, a main component of tight junctions, is discussed for the leak pathway [20,21]. Finally, the tricellular junction (structurally distinct tight junctions formed at the contact points of three cells) was identified as a potential pathway of macromolecule passage [22,23,24].

In this study, we quantified macromolecule translocation across mucosal samples obtained from HIV-infected patients (treatment-naive and patients treated with combination antiretroviral therapy (cART)) and HIV-negative control persons. Our data indicates the increased macromolecule permeability in the intestinal mucosa of untreated HIV-infected patients whereas the mucosal macromolecule permeability of patients treated by cART was not different from that of the uninfected controls. Combining translocation studies on mucosal samples with mechanistic studies in a cell culture model, we identified two distinct pathways of increased macromolecule translocation in HIV infection: transcytosis and passage through apoptotic leaks. 

## 2. Materials and Methods

### 2.1. Patients

The study was approved by the local ethics committee (Ethics Committee of Charité—Universitätsmedizin Berlin, no. EA4/015/13) and all patients gave written informed consent before study participation. Patients included underwent routine endoscopy for diagnostic evaluation of gastrointestinal symptoms, for ruling out neoplastic disease, or for unexplained anemia. Patients with intestinal infections, diarrhea, inflammatory bowel disease, or a pathological lesion of the intestinal mucosa as demonstrated by endoscopy or histology were excluded. Mucosal samples were obtained from three patient groups: treatment-naive HIV-infected patients, HIV-infected patients treated by combination antiretroviral therapy (cART), and HIV-negative patients (control group). Patient characteristics are provided in Table 1. Treatment-naive HIV-infected patients had not received antiretroviral therapy before the time of the mucosal sampling. In all treated patients, viral replication has been continuously suppressed below the limit of detection (quantitative polymerase chain reaction <20 copies/mL, Roche Amplicor, Roche, Mannheim, Germany) for at least 18 months. In this respect, a viral load higher than 20 cop/mL (limit of detection) but lower than 400 cop/mL was not considered a break of viral suppression, if a subsequent viral load was undetectable again. cART consisted of a standard regimen of two nucleoside reverse transcriptase inhibitors combined with either a ritonavir-boosted protease inhibitor, an integrase inhibitor, or a non-nucleoside reverse transcriptase inhibitor. 

Experiments were performed as described in the following sections. As for some patients, several samples were available, and the number of performed experiments (m) may differ from number of the individuals analyzed (*n*). For statistics, the number of individuals was used.

### 2.2. Macromolecule Permeability of the Duodenal and Sigmoid Mucosa

Mucosal specimens were obtained from the duodenal or sigmoid mucosa with 3.4 mm biopsy forceps. The disc-mounted mucosa was placed into modified Ussing-type chambers as described [25]. Chemical gradients were eliminated by filling both the mucosal and the serosal compartment with the same modified Ringer’s solution [25]. The solution was gassed with 95% O_2_/5% CO_2_, and the pH was 7.4. The temperature was maintained at 37 °C using water-jacketed reservoirs. For cold inhibition experiments, permeabilities were determined in mucosal samples obtained from the same patients in parallel at 37 °C and 14 °C. To neutralize the effects of electrical gradients, flux experiments were performed under voltage-clamp conditions. Mucosal integrity was monitored using continuous measurements of the transepithelial resistance during the flux experiments.

Fluorescein isothiocyanate (FITC)-labeled dextran 4000 (FD4, MW 4 kDa; TdB Labs, Uppsala, Sweden) and horseradish peroxidase (HRP, MW 45 kDa; Sigma Aldrich, Traufkirchen, Germany) were used as macromolecular markers. Macromolecule permeabilities were calculated from mucosal to serosal fluxes obtained in standardized flux experiments. For FD4 fluxes, 0.4 mM of pre-dialyzed FD4 was applied to the mucosal and 0.4 mM of unlabeled dextran 4 kDa (Serva, Heidelberg, Germany) to the serosal compartment. Serosal samples were taken at 0, 30, 60, 90, and 120 min after application and FD4 concentrations were determined using a photometer at 520 nm (Tecan Infinite M200, Tecan, Männedorf, Switzerland). For measuring HRP fluxes, HRP was added to the mucosal side to a final concentration of 20 µM. HRP concentration in the serosal compartment was quantified with a fluorogenic peroxidase substrate kit (Quanta Blu™, Thermo Fisher Scientific, Darmstadt, Germany). Permeabilities were calculated as the ratio of flux J (mol·h^−1^·cm^−2^) over concentration gradient Δc (mol/L).

### 2.3. Cell Culture Experiments

T84 cells were cultured and seeded onto Millicell PCF filters (effective area: 0.6 cm^2^, Millicell PCF, Millipore, Schwalbach, Germany) as described [26]. As confirmed by measurement of transepithelial resistance [26], the monolayers reached stable confluence after an average of nine days after seeding. The permeability of the monolayers to FD4 and HRP was determined in six-well plates with a glucose-enriched (10 mM) HEPES-buffered Ringer in both the apical and basolateral compartments. FD4 and HRP permeabilities were determined with the protocol described for native epithelia. Control experiments performed in six-well plates and Ussing chambers in parallel yielded qualitatively identical macromolecule permeabilities. 

### 2.4. Apoptosis Assays and Inhibitor Studies

Apoptosis was induced by 24 h incubation with 1 µg/mL camptothecin (Sigma Aldrich) added on both sides of the monolayers. For inhibition experiments, the apoptosis inhibitor Q-VD-Oph (20 µM; Merck, Darmstadt, Germany) or the endocytosis inhibitor Myristyltrimethylammoniumbromid (20 µM, MiTMAB, Calbiochem, Leiden, The Netherlands) was added 2 h prior to camptothecin. Epithelial apoptosis was quantified by measurement of the relative caspase activity with a commercial assay (SensoLyte^TM^ Homogeneous AFC Caspase-3/7 Assay Kit, Anaspec, Fremont, CA, USA). 

In tissue sections, epithelial apoptosis was quantified by TUNEL staining, which was performed as described before [25] and analyzed using the TUNEL cell counter 3.0 plugin [27] of FIJI with free-hand selection ROI evaluating TUNEL-positive nuclei and DAPI-stained nuclei. In addition, apoptosis was also analyzed by Western Blot (see below) using antibodies against cleaved Caspase-3 (rabbit anti-cleaved Caspase-3, Cell Signaling, Leiden, The Netherlands, Cat.-No. #9661).

### 2.5. Western Blots

For analysis of the tight junction composition, membrane protein fractions were prepared [23], and the expression of tight junction proteins discussed as being involved in macromolecule pathway regulation was quantified by immunoblotting employing primary antibodies against tricellulin (rabbit anti-Tric, Invitrogen, Dreireich, Germany, Cat.-No. 700191), occludin (rabbit anti-occludin, Invitrogen, Cat.-No. 711500), angulin-1 (rabbit anti-LSR, Atlas Antibodies, Bromma, Sweden, Cat.-No. HPA007270), and β-Actin (mouse anti-b-Actin, Sigma Aldrich, Cat.-No. A5441) as described [28]. Specific signals were quantified by chemiluminescence detection after incubation of the washed membranes with Lumi-Light plus (Hoffman La Roche, Basel, Switzerland) using luminescence imaging (Fusion FX7, Vilber Lourmat, Eberhardzell, Germany) and quantification software (Multi-Gauge V2.3, FujiFilm, Tokyo, Japan).

### 2.6. Statistical Analysis

Results are given as means ± SEM. In univariate analysis, significance was tested by the 2-tailed Student’s *t*-test. Multivariate analysis was performed using the one-way ANOVA. Bonferroni–Holm adjustment was used for post hoc analysis in multiple testing. *p*-values <0.05 were considered significant.

## 3. Results

### 3.1. Increased Macromolecule Translocation in Untreated HIV-Infection 

In the duodenal mucosa of treatment-naive HIV-infected patients, the macromolecular permeability was almost fivefold increased over that of HIV-negative controls (Figure 1A,B; FD4: 4.5 ± 1.5 × 10^−6^ cm/s versus 0.9 ± 0.2 × 10^−6^ cm/s, *p* = 0.04; HRP: 69.1 ± 17.1 × 10^−6^ cm/s versus 17.3 ± 3.61 × 10^−6^ cm/s, *p* = 0.007; untreated HIV-infected patients versus HIV-negative controls, respectively). Also, in the colon mucosa of untreated patients, FD4 and HRP permeabilities were numerically higher than in the mucosa from HIV-negative controls, but the differences observed were not statistically significant (Figure 1D,E; FD4: 4.0 ± 1.9 × 10^−6^ cm/s versus 0.9 ± 0.3 × 10^−6^ cm/s, *p* = 0.08; HRP: 33.1 ± 27.8 × 10^−6^ cm/s versus 6.1 ± 2.9 × 10^−6^ cm/s, *p* = 0.37). In contrast, in HIV-infected patients treated with cART, the duodenal HRP and FD4 permeabilities were not different from those of uninfected controls (Figure 1A,B,D,E). Thus, the increased macromolecule permeability of the duodenal mucosa of untreated HIV-infected patients is linked to uncontrolled viral replication.

After having established that untreated HIV infection is associated with increased macromolecule permeability of the intestinal mucosa, we tried to define the route of macromolecule translocation. To check for the presence of large-area mucosal lesions, we continuously monitored the transepithelial resistance (TER) which can be taken as a rough proxy for the overall integrity of the mucosa. As we did not observe any differences in TER in the duodenal or colon mucosa obtained from the HIV-infected patients and HIV-negative controls (Figure 1C,F), there was no evidence of gross disruptions of the mucosal lining such as erosions or ulcers. We concluded that the HIV-induced macromolecule translocation was caused by more subtle mechanisms across a largely intact intestinal epithelium. Three pathways were specifically addressed: (i) the paracellular route across the tricellular junction, (ii) macromolecule diffusion through apoptotic leaks, and (iii) active transcellular macromolecule transportation by transcytosis.

### 3.2. The Paracellular Pathway as a Potential Route for Macromolecule Translocation

During the last decade, the tricellular junction was recognized as a pathway of paracellular macromolecule translocation. Specifically, the tight junction protein tricellulin has been shown to seal the tricellular junction against macromolecule passage [23]. Accordingly, reduced expression of tricellulin, as well as reduced expression of occludin and the lipolysis-stimulated lipoprotein receptor (LSR, also termed angulin-1), both of which mediate the tricellular localization of tricellulin, have been linked to increased permeability of the tricellular junction [23,29,30].

We did not observe a decrease but rather an increased expression of tricellulin, occludin, and LSR in the duodenal mucosa (Figure 2A,C) of HIV-infected patients compared to the HIV-negative controls. In addition, there were no differences in the expression of these tight junction proteins in the colon mucosa (Figure 2B,D). Thus, we did not find any evidence for increased macromolecule passage across the tricellular junction in HIV-infected patients, but hypothesize that the upregulations observed could be of counter-regulatory nature.

### 3.3. Contribution of Enterocyte Apoptosis to Macromolecule Translocation

Using DNA fragmentation for quantification of apoptosis (TUNEL assay, see Section 2), we found increased epithelial apoptotic rates in both duodenal and colon mucosa in untreated HIV-infected patients compared to the HIV-negative controls (Table 2). In addition, the presence of cleaved Caspase-3 was increased in the mucosa of untreated HIV-infected patients (Appendix A).

Furthermore, in the TUNEL assay, the mucosal apoptotic rates of treated HIV-infected patients were not different from that of the HIV-negative controls (Table 2). Thus, the epithelial apoptotic rate and mucosal macromolecule permeability were altered accordingly in untreated and treated patients, respectively. Therefore, we hypothesized that enterocyte apoptosis might represent a mechanism of the observed macromolecule translocation.

To analyze whether or not enterocyte apoptosis can be sites of macromolecule translocation, we performed flux experiments in T84 monolayers. Due to the required duration of incubation with inducers or inhibitors, these experiments were not feasible in the intestinal biopsies because of their time-constrained survival in the Ussing chamber. T84 cells are a widely used in vitro model for the analysis of the intestinal epithelial barrier. Employing this model epithelium enabled us to pharmacologically modulate enterocyte apoptosis. As expected, the apoptosis inducer camptothecin induced apoptosis in T84 cells, which was reflected by a significant drop in TER (Figure 3A) and increased relative caspase activity (Figure 3B). Furthermore, camptothecin also strongly increased the permeability of the monolayers to FD4 and HRP at both 37 °C and 14 °C (Figure 3C,D). 

Conversely, a pan-caspase inhibitor (Quinoline-Val-Asp-Difluorophenoxymethylketone, QVD-Oph) effectively inhibited the camptothecin-induced macromolecule permeabilities to FD4 and HRP (Figure 4A,B). In conclusion, epithelial apoptosis induced a temperature-independent macromolecule translocation across monolayers of T84 cells, and inhibiting epithelial apoptosis inhibited the macromolecule translocation induced by camptothecin. Taken together, these results indicate a straightforward mechanistic link between enterocyte apoptosis and macromolecule translocation.

### 3.4. Macromolecule Translocation by Transcytosis 

In contrast to passage via apoptotic leaks, which is a passive diffusional process driven by concentration gradients, transcytosis relies on endocytic uptake into the enterocytes and directed vesicular transport within the epithelial cells.

To test whether transcytosis contributes to FD4 and HRP translocation, we inhibited endocytosis in T84 cells treated with or without camptothecin by the addition of MitMAB, which ameliorated only the increase in camptothecin-induced HRP but not FD4 permeability (Figure 4C,D). Furthermore, we performed flux experiments in T84 monolayers at low temperatures and compared the macromolecule permeabilities for FD4 and HRP. Transcytosis is an energy-dependent active-cellular process that can be inhibited by cold. On the other hand, passive translocation mechanisms driven by diffusion gradients are not sensitive to cold. Again, HRP permeability was inhibited by cold (Figure 3D) while FD4 permeability was not affected (Figure 3C). Taken together, these data support the assumption that FD4 translocation is a diffusional process occurring along paracellular pathways, whereas HRP is also actively transported across the epithelial monolayers by transcytosis. 

In a similar way, we compared the permeabilities for FD4 and HRP in duodenal and colon mucosa at 14 °C and 37 °C. As shown in Figure 5A, the permeability to FD4 was unaffected by cold which is well compatible with passive, diffusional permeation of FD4 through apoptotic leaks or other paracellular pathways. On the other hand, the permeability of the duodenal mucosa to HRP was reduced at 14 °C by almost 70% (37 °C and 14 °C: 10.0 ± 2.6 × 10^−9^ cm/s versus 3.2 ± 1.0 × 10^−9^ cm/s, respectively, *p* = 0.03; Figure 5B). Thus, these data imply that transcytosis is involved in HRP but not FD4 translocation across the small intestinal mucosa. 

## 4. Discussion

Increased microbial translocation has been proposed as a driver of disease progression and comorbidity in HIV infection for many years, although this conception was hitherto based mainly on the finding of increased levels of microbial macromolecules in the blood of untreated HIV-infected patients [1,2,4,5,6,9,25,31]. However, as the systemic concentration of gut-derived microbial components is determined not only by the amount of mucosal translocation but also by the clearing capacity of the immune system [16], raised systemic levels of microbial macromolecules per se do not prove the presence of a macromolecular mucosal barrier defect. Using a quantitative assessment of mucosal macromolecule translocation, the present study, for the first time, unequivocally demonstrates an increased macromolecule permeability of the intestinal mucosa of HIV-infected patients. According to our data, the macromolecule permeability of the intestinal mucosa is markedly increased in patients with an untreated HIV infection but returns to normal values in patients effectively treated by combination antiretroviral therapy. We consider these data the strongest evidence so far for the conception that the raised serum levels of microbial macromolecules found in patients with untreated HIV infection are secondary to increased mucosal macromolecule translocation.

These results represent a substantial expansion of previous studies reporting increased mucosal permeability to ions and small molecules in HIV-infected patients [25,31]. As macromolecules are thought to translocate via specific pathways [17,18], the mechanisms of ions and small-size molecules identified in previous studies cannot be assumed to also be responsible for the increased translocation of macromolecules. In fact, little is known about the mechanisms of increased macromolecule translocation in HIV infection. Authors commenting on the pathways of macromolecule translocation in HIV infection usually refer to either increased serum levels of so-called markers of gut mucosal dysfunction, such as E-cadherin, or altered mucosal expression of certain tight junction proteins (for review see [32]). However, whether or not these alterations are causally linked to a pathological macromolecule translocation is unknown, because there are no data functionally linking the analyzed markers and proteins to the macromolecule permeability of the intestinal mucosa. To overcome this limitation, we based our efforts to identify the pathways of mucosal macromolecule translocation on a quantitative assessment of mucosal macromolecule permeability.

In general, macromolecules can travel across the epithelium along two translocation routes: the transcellular and the paracellular pathway. Transcellular translocation of macromolecules relies on active cellular transcytosis that can be reduced or even inhibited by cold [33] of specific inhibitors. Paracellular translocation, on the other hand, is ruled by the laws of diffusion and, therefore, not inhibited at 14 °C. The cold inhibition experiments performed in mucosal samples indicate that the bulk of HRP translocation occurs via transcytosis whereas that of FD4 follows a paracellular pathway. Accordingly, the cell culture experiments showed that HRP, but not FD4 translocation, was strongly reduced by cold as well as by pharmaceutical inhibition of endocytosis. Similar to these findings, animal studies and studies in patients with Crohn’s disease demonstrated mucosal translocation of large macromolecules such as HRP (MW 44 kDa), ovalbumin (MW 43 kDa), and LPS (MW appr. 90 kDa) by transcytosis [33,34,35]. Based on these data, it seems plausible that the increased transcytosis of large macromolecules represents major mechanisms of mucosal macromolecule translocation in untreated HIV infection.

In contrast to HRP translocation, FD4 translocation was neither inhibited by cold nor by endocytosis inhibitors. This indicates that paracellular pathways permissive to diffusional translocation of macromolecules are also activated in untreated HIV-infected patients. Candidate sites for paracellular macromolecule passage in HIV infections are the tricellular junction and epithelial leaks resulting from enterocyte apoptosis [9,13,23,25]. The macromolecular permeability of the tricellular junction has been functionally linked to the expression of proteins that seal the tricellular junction against macromolecule passage, such as tricellulin, occludin, and the lipolysis-stimulated lipoprotein receptor (LSR) also known as angulin-1 [23,29,30]. However, we did not find a reduced but rather an increased mucosal expression of these sealing proteins in our group of untreated HIV-infected patients. Therefore, we assume an HIV-induced increase in macromolecule passage across the tricellular junction is unlikely. The observed upregulations might be interpreted as resulting from counter-regulatory processes in response to increased macromolecule passage via apoptotic leaks.

In good agreement with previous studies [9,13,25,36], we observed an increased rate of epithelial apoptosis in our group of untreated HIV-infected patients. However, the factual contribution of enterocyte apoptosis to macromolecule translocation in HIV infection has been unclear, because, to a certain degree, the intestinal epithelium is able to maintain the epithelial barrier during the shedding of apoptotic enterocytes by a plastic rearrangement of surrounding cells [9,37]. Nevertheless, increased apoptotic rates in HIV patients were correlated with the increased presence of surrogate markers for macromolecule passage (sCD14 and IDO-1 activity) already suggesting that this potentially might be a possible and important pathway [9]. Our results now provide several lines of evidence for a significant contribution of enterocyte apoptosis to macromolecule translocation in HIV-infected patients. First, the increased enterocyte apoptosis was associated with increased translocation of FD4. Second, the FD4 translocation was insensitive to cold which, as outlined before, is well compatible with diffusion through apoptotic leaks within the epithelium. Third, the increased apoptotic rate and FD4 permeability were both restored to control levels in the duodenal mucosa of patients treated by cART. Thus, without increased epithelial apoptosis, increased FD4 translocation did not occur. Finally, in our cell culture model, experimental induction of apoptosis enhanced macromolecule translocation, and inhibition of apoptosis reduced both enterocyte apoptosis as well as macromolecule translocation. Therefore, we propose that passage at sites of enterocyte apoptosis should be regarded as an important pathway of macromolecule translocation in HIV infection. The question remains as to why previous studies failed to find an effect of enterocyte apoptosis on macromolecule translocation [37]. Possibly, the apoptosis pathway comes into action only, if compensatory mechanisms maintaining the epithelial continuity during the shedding of apoptotic enterocytes are inhibited or stretched beyond their capacity by a chronically increased rate of epithelial apoptosis. 

## 5. Conclusions

In summary, based on a quantitative assessment of macromolecule translocation across area-defined explants of the intestinal mucosa, our study demonstrates an increased mucosal macromolecule permeability in untreated but not in treated HIV-infected patients. We identified two pathways of macromolecule translocation to be activated in untreated HIV infection. While some macromolecules (such as HRP) are actively taken up by transcytosis, other macromolecules (such as FD4) are not transcytosis and exclusively diffuse along the paracellular pathway. The increased occurrence of mucosal apoptotic leaks in untreated HIV infection increases this uptake. Additionally, parts of the transcytotically passing substances can diffuse through the apoptotic leaks as well. These mechanisms of macromolecule translocation identified in untreated HIV-infected patients probably also apply to non-infectious diseases associated with increased serum concentration of microbial macromolecules and increased enterocyte apoptosis such as inflammatory bowel disease, Whipple’s disease, or celiac sprue.

## Figures and Tables

**Figure 1 cells-12-01887-f001:**
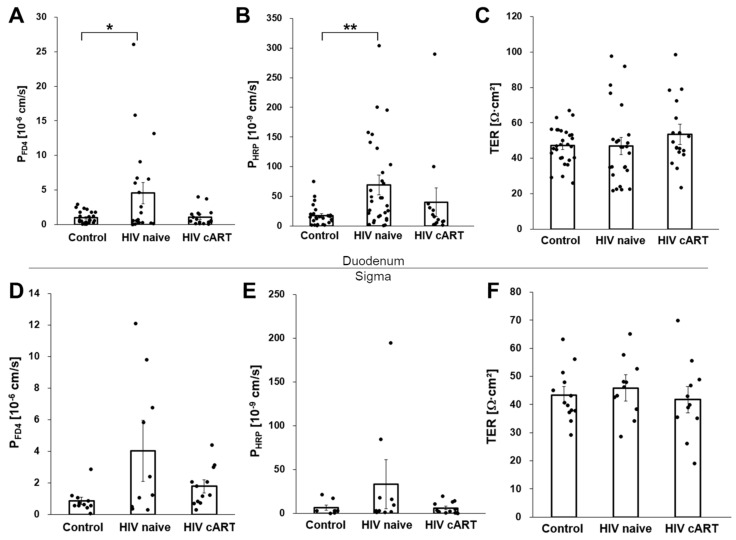
Macromolecule permeabilities in the duodenal and colon mucosa of HIV-infected patients and HIV-negative controls. Permeabilities were determined from FD4 and HRP flux measurements performed on duodenal (**A**,**B**) and colon (**D**,**E**) mucosa obtained from HIV-negative control individuals, from untreated (HIV naive), and from suppressively treated (HIV cART) HIV-infected patients. The transepithelial resistance (TER) of the mucosal samples was monitored in parallel to the flux measurements (**C**,**F**). m = 25–28, *n* = 20 (control), m = 23–32, *n* = 19 (HIV), m = 14–18, *n* = 10 (HIV cART), and m = 8–13, *n* = 10 (control), m = 10–11, *n* = 5 (HIV), m = 11–12, *n* = 9 (cART) for duodenal and colon mucosal samples, respectively. * *p* < 0.05, ** *p* < 0.01.

**Figure 2 cells-12-01887-f002:**
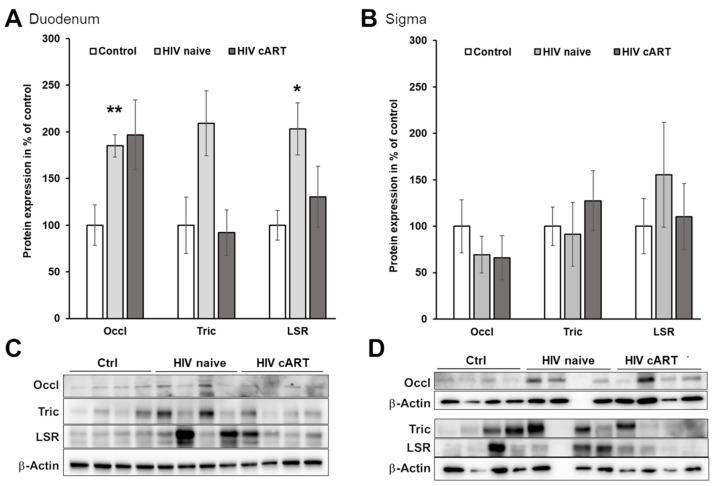
Expression of tight junction proteins involved in paracellular macromolecule passage. For comparison, protein expression was expressed in relation to that of controls (100%). (**A**) Expression of occludin (Occl), tricellulin (Tric), and lipolysis-stimulated lipoprotein receptor (LSR) in duodenal mucosa obtained from HIV-negative control individuals (*n* = 6), and from untreated (HIV naive; *n* = 10) and suppressively treated (HIV cART; *n* = 7) HIV-infected patients. (**B**) Expression of occludin, tricellulin, and LSR in colon mucosa obtained from HIV-negative control individuals (*n* = 4), and from untreated (HIV naive; *n* = 3) and suppressively treated HIV-infected patients (HIV cART; *n* = 3–4). Representative immunoblots of occludin, tricellulin, and LSR in membrane preparations from duodenal (**C**,**D**) colon mucosa. The band covered with background at the very right position of the blot was excluded from the densitometric analysis for tricellulin. * *p* < 0.05, ** *p* < 0.01.

**Figure 3 cells-12-01887-f003:**
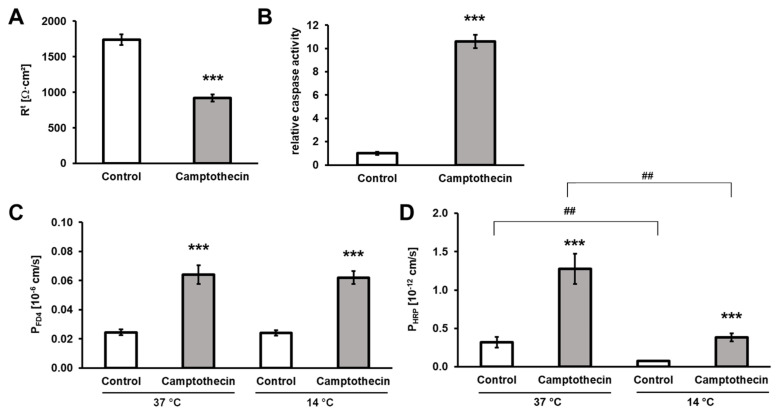
Effect of apoptosis induction and cold on macromolecule permeability of T84 monolayers. Induction of apoptosis by camptothecin (1 µg/mL) (**A**) reduced the TER of T84 cells (*n* = 13) and (**B**) increased the activity of caspases (*n* = 4; for comparison, the caspase activity of camptothecin-treated monolayers was expressed in relation to that of untreated control monolayers). It also affected permeability of T84 monolayers at 37 °C and 14 °C to (**C**) FD4 and (**D**) HRP. *n* = 13 and 10 for FD4 and HRP fluxes at 37 °C, respectively, and *n* = 10 and 7 for FD4 and HRP fluxes at 14 °C, respectively. *** *p* < 0.001, camptothecin versus control. ## *p* < 0.01, 37 °C versus 14 °C.

**Figure 4 cells-12-01887-f004:**
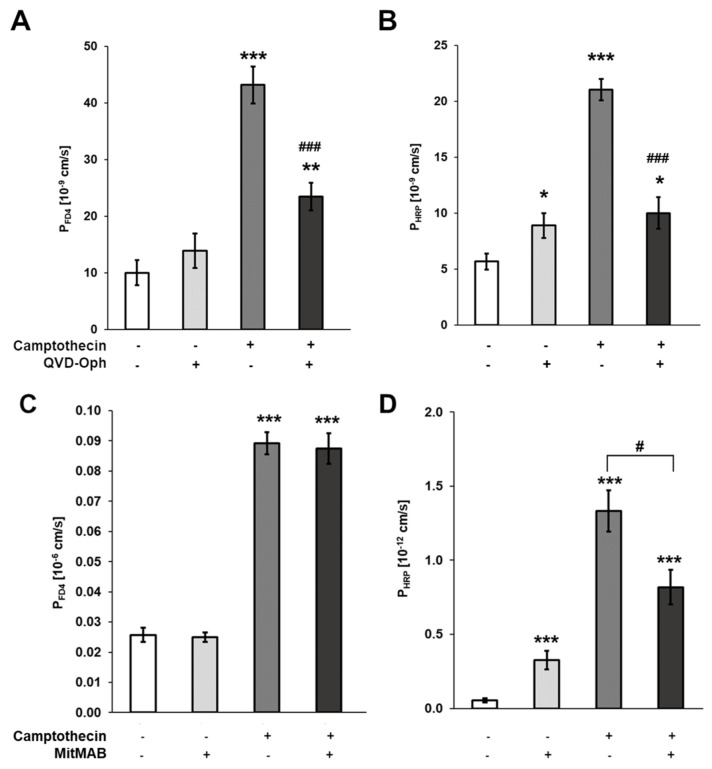
Effect of inhibitors of apoptosis or endocytosis on camptothecin-induced macromolecule permeability of T84 monolayers. Effect of camptothecin (1 µg/mL) on permeability of T84 monolayers to FD4 and HRP in the absence or presence of the caspase inhibitor QVD-Oph (20 µM) (**A**,**B**), or the endocytosis inhibitor MitMAB (20 µM) (**C**,**D**). * *p* < 0.05, ** *p* < 0.01, *** *p* < 0.001 versus untreated controls. # *p* < 0.05, ### *p* < 0.001 without versus with inhibitor. *n* = 7–9.

**Figure 5 cells-12-01887-f005:**
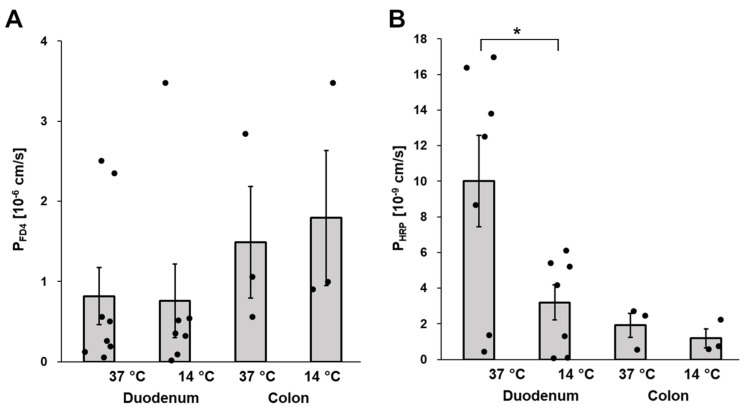
Effect of cold on macromolecule permeability of the duodenal and colon mucosa. Comparison of (**A**) FD4 and (**B**) HRP permeabilities of the duodenal and the colon mucosa at 37 °C and 14 °C. Mucosal samples for these experiments were obtained from HIV-negative individuals. * *p* < 0.05, n = 7–8 for duodenal samples, *n* = 3 for colon samples.

**Table 1 cells-12-01887-t001:** Patient characteristics. Values are given as absolute numbers or as means ± SEM. ND: not determined, NA: not applicable, <LOD = below limit of detection (<20 copies/mL).

	Controls	HIV Naive	HIV cART
Duodenum			
N	20	19	10
Sex (m/w)	10/10	14/5	8/1
Age (years)	49.5 (±2.3)	43.5 (±2.5)	57.1 (±1.7)
CD4+ T cells (cells/µL)	ND	239 (±20)	371 (±1.5)
Viral load (log10 copies/mL)	NA	5.3 (±2.5)	<LOD
Colon			
N	10	5	9
Sex (m/w)	7/6	4/1	9/0
Age (years)	51.3 (±2.7)	31.5 (±2.4)	64.6 (±0.9)
CD4+ T cells (cells/µL)	ND	130 (±13)	563 (±12)
Viral load (log10 copies/mL)	NA	5.2 (±2.7)	<LOD

**Table 2 cells-12-01887-t002:** Epithelial apoptosis in the duodenal and colon mucosa of HIV-infected patients and HIV-negative controls. Epithelial apoptosis was quantified as percent of apoptotic nuclei identified by TUNEL staining in HIV-negative control individuals (ctrl), untreated (HIV naive), and suppressively treated (HIV cART) HIV-infected patients. n represents the number of analyzed patients, m is the number of analyzed nuclei.

	*n*	m	Apoptosis %	SEM	*p* (to Ctrl)	*p* (to HIV)
**Duodenum**						
**Ctrl**	14	33,990	1.52	0.23	-	-
**HIV naive**	10	24,458	3.25	0.70	0.0278	-
**HIV cART**	10	21,357	2.07	0.25	0.1249	0.1298
**Colon**						
**Ctrl**	20	37,391	2.11	0.43	-	-
**HIV naive**	19	48,837	4.13	0.68	0.0310	-
**HIV cART**	17	39,882	1.87	0.47	0.7085	0.0115

## Data Availability

The data presented in this study are also available on request from the corresponding author.

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
