# Peer review of "Macromolecule Translocation across the Intestinal Mucosa of HIV-Infected Patients by Transcytosis and through Apoptotic Leaks"

_cells, 2023, doi:10.3390/cells12141887_

Round 1

Reviewer 1 Report

The is a nicely written paper that makes the important point that molecular translocation at the gut barrier with HIV infection, specifically the mechanisms for it, are not clear.  

There is a nice discussion of the problem studied, its clear and the methods and results clear.  There is some minor confusion reading the abstract with regard to HIV and HIV and ART versus aRT naive individuals.

Figures 1 and 2 are pretty clear and convincing.  The remaining figures try to study the role of apoptosis of epithelial cells in translocation.  The data are convincing that apoptosis occurred using the monocyte layer cells and can be blocked, but it is not clear how translatable this is to the tissues/barriers in HIV gut, as demonstrated in data from Figures 1 and 2.  

The authors do not provide a rationale for using the T84 cells with regard to HIV gut and gut tissue culture models.  It seems this should be addressed for the experiments to be appreciated in the context of HIV.

Reviewer 2 Report

In the submitted manuscript, the authors investigated translocation of macromolecules across intestinal mucosa of HIV-infected patients. The study is interesting and data are well-presented.

Reviewer 3 Report

Susanne M. Krug et colleagues have conducted a meticulous investigation into the permeability of the intestinal mucosa in HIV-infected patients and have shed light on the mechanisms of macromolecule translocation having in mind previous studies and suggestions regarding increased macromolecule translocation [Ref: 7, 9-15]. This research has significant implications for the understanding and treatment of HIV infection due to the comprehensive approach and careful execution of experiments. I commend the authors for their meticulous work and their contribution to our understanding of macromolecule translocation across the intestinal mucosa in HIV-infected patients. The title accurately reflects the study's focus and findings, emphasizing the two main pathways involved in macromolecule translocation: transcytosis and diffusion through apoptotic leaks. This research provides valuable insights into the underlying mechanisms of mucosal barrier dysfunction in HIV infection and its broader relevance to diseases characterized by increased serum levels of microbial macromolecules and enterocyte apoptosis.

Specific Comments about the study

-The strength of this study lies in its quantitative assessment of macromolecule translocation, which was performed using area-defined explants of intestinal mucosa. This approach enabled the authors to provide evidence of increased mucosal macromolecule permeability in untreated HIV-infected patients. By differentiating between transcellular and paracellular pathways, the authors have advanced our understanding of the complex mechanisms involved in macromolecule translocation.

-The findings regarding the activation of both transcytosis and diffusion through apoptotic leaks provide valuable insights into the routes of macromolecule translocation in HIV infection. The observed association between increased enterocyte apoptosis and higher permeability to certain macromolecules underscores the significance of apoptosis as a contributing factor. The efforts to elucidate the role of enterocyte apoptosis through experimental induction in a cell culture model further strengthen this observation.

-Furthermore, this study has important implications for the management of HIV-infected patients. The observed restoration of mucosal macromolecule permeability to normal values in patients effectively treated with combination antiretroviral therapy highlights the potential of therapy in restoring mucosal barrier function. These findings highlight the importance of timely initiation of treatment and its potential impact on preventing disease progression and comorbidities associated with HIV infection.

Round 2

Reviewer 1 Report

Figures 1 and 2 raise important points.  The other Figures relative to apoptosis make a different point.  If the authors were to demonstrate apoptosis in gut tissue sections of their studies or minimal cite it is found in HIV infection with disease and translocation, the data would be meaningful.  At present, it is merely and only suggestive. 

Author Response

Increased apoptotic rates have been reported for HIV patients before, which we also mention in the text, for example citation 9 and 31. In the work of Sonsouk et al. (citation 9) these were also correlated to the increase in surrogate markers for marcomolecule passage.

To make this clearer, we added a sentence to the discussion pointing it out in more detail than before; line 380: "Nevertheless, increased apoptotic rates in HIV patients were correlated with increased presence of surrogate markers for macromolecule passage (sCD14 and IDO-1 activity) already suggesting that this potentially might be a possible and important pathway [9]."